# Zfra Inhibits the TRAPPC6AΔ-Initiated Pathway of Neurodegeneration

**DOI:** 10.3390/ijms232314510

**Published:** 2022-11-22

**Authors:** Yu-Hao Lin, Yao-Hsiang Shih, Ye Vone Yap, Yen-Wei Chen, Hsiang-Lin Kuo, Tsung-Yun Liu, Li-Jin Hsu, Yu-Min Kuo, Nan-Shan Chang

**Affiliations:** 1Institute of Molecular Medicine, College of Medicine, National Cheng Kung University, Tainan 70101, Taiwan; 2Department of Cell Biology and Anatomy, College of Medicine, National Cheng Kung University, Tainan 70101, Taiwan; 3Department of Anatomy, School of Medicine, College of Medicine, Kaohsiung Medical University, 100, Shih-Chuan 1st Road, Sanmin District, Kaohsiung 80708, Taiwan; 4Department of Medical Laboratory Science and Biotechnology, College of Medicine, National Cheng Kung University, Tainan 70101, Taiwan; 5Department of Neurochemistry, New York State Institute for Basic Research in Developmental Disabilities, New York, NY 10314, USA; 6Graduate Institute of Biomedical Sciences, College of Medicine, China Medical University, Taichung 404333, Taiwan

**Keywords:** tumor suppressor, p53, WWOX, TRAPPC6A, TRAPPC6AΔ, TIAF1, SH3GLB2, Alzheimer’s disease

## Abstract

When WWOX is downregulated in middle age, aggregation of a protein cascade, including TRAPPC6AΔ (TPC6AΔ), TIAF1, and SH3GLB2, may start to occur, and the event lasts more than 30 years, which results in amyloid precursor protein (APP) degradation, amyloid beta (Aβ) generation, and neurodegeneration, as shown in Alzheimer’s disease (AD). Here, by treating neuroblastoma SK-N-SH cells with neurotoxin MPP+, upregulation and aggregation of TPC6AΔ, along with aggregation of TIAF1, SH3GLB2, Aβ, and tau, occurred. MPP+ is an inducer of Parkinson’s disease (PD), suggesting that TPC6AΔ is a common initiator for AD and PD pathogenesis. Zfra, a 31-amino-acid zinc finger-like WWOX-binding protein, is known to restore memory deficits in 9-month-old triple-transgenic (3xTg) mice by blocking the aggregation of TPC6AΔ, SH3GLB2, tau, and amyloid β, as well as inflammatory NF-κB activation. The Zfra4-10 peptide exerted a strong potency in preventing memory loss during the aging of 3-month-old 3xTg mice up to 9 months, as determined by a novel object recognition task (ORT) and Morris water maize analysis. Compared to age-matched wild type mice, 11-month-old *Wwox* heterozygous mice exhibited memory loss, and this correlates with pT12-WWOX aggregation in the cortex. Together, aggregation of pT12-WWOX may link to TPC6AΔ aggregation for AD progression, with TPC6AΔ aggregation being a common initiator for AD and PD progression.

## 1. Introduction

Neurodegeneration and related dementia normally occur in humans after 65 years of age [1,2,3,4,5]. Many neurodegenerative diseases such as Parkinson’s disease (PD), Alzheimer’s disease (AD), and Huntington’s disease possess neuronal protein aggregates or inclusion bodies as one of the pathological hallmarks of the diseases [1,2,3,4,5]. Indeed, the proteins may start to aggregate and accumulate in the brain of middle-aged normal humans beginning at 40 years old [1,2,3,4,5]. AD is a chronic neurodegenerative disease that usually starts slowly from an asymptomatic stage and becomes worse as one grows older. The most common early symptom is difficulty in remembering recent events (short-term memory loss) [1,2,3,4,5]. As the disease advances, AD patients may suffer problems associated with language, disorientation, mood swings, and loss of motivation.

WW-domain-containing oxidoreductase (WWOX) is generally considered as an untypical tumor suppressor [6,7,8,9,10,11,12]. WWOX participates in cell death, differentiation, signaling, and protein–protein interactions, as well as in cell migration and recognition [10,11,12,13,14,15,16,17,18,19,20,21,22]. Substantial evidence has shown that WWOX plays a critical role in limiting neurodegeneration such as AD and neuronal damage [23,24,25,26,27,28,29,30,31,32,33,34]. Loss of WWOX in newborns results in severe neural diseases, metabolic disorders, mental retardation, stunt growth, and early death [28,35,36,37,38,39,40]. Most recently, the *WWOX* gene has been defined as a risk factor for AD, on the basis of established in vitro and in vivo studies for more than 20 years, along with the recent genome wide analysis [41]. WWOX possesses two *N*-terminal WW domains: a nuclear localization signal in between the WW domains and a *C*-terminal short-chain alcohol dehydrogenase/reductase (SDR) domain [6,7,8,9]. A mitochondrial localization region is located in the SDR domain [6,7,8,9]. The first WW domain (WW1) with Tyr33 phosphorylation binds JNK and ERK and thereby prevents Tau hyperphosphorylation by these enzymes [23]. The SDR domain binds GSK3β and the C-terminal region of Tau. This binding leads to inhibition of Tau hyperphosphorylation [23]. 

Downregulation of WWOX in the brain may start to occur in middle age, and this leads to spontaneous aggregation of a cascade of intracellular proteins, including TIAF1 (TGFβ-induced antiapoptotic protein) [42,43], TRAPPC6AΔ (trafficking protein particle complex 6A isoform, TPC6AΔ) [44,45,46,47,48], and SH3GLB2 (SH3-domain-containing GRB2-like, endophilin B) [5,49,50]. Compared to the wild type, the TPC6AΔ isoform has an internal deletion of 14 amino acids in the *N*-terminus [44]. The protein aggregates are deposited in the mitochondria and continuously induce caspase activation, leading to degradation of amyloid precursor protein (APP), generation of Aβ, and eventual cell death [43,44]. 

While WWOX is downregulated in middle age, TPC6AΔ, together with downstream TIAF1, SH3GLB2, APP, amyloid beta (Aβ), and tau, starts to aggregate in a cascade-like manner [42,43,44,45,46,47]. Here, we showed that neurotoxin-induced death of neuroblastoma cells is associated with increased aggregation of these proteins. A naturally occurring small-zinc-finger-like peptide that regulates apoptosis, designated Zfra, mitigates the AD-like symptoms in triple-transgenic mice and the growth of many cancer cells [5,51,52,53,54]. Zfra activates a novel Hyal-2+ Z lymphoid cell to kill cancer cells [5,51,52,53,54]. Whether Z cells prevent the progression of AD-like symptoms in mice is unknown. Here, we determined the potent efficacy of Zfra in preventing the progression of AD-like symptoms during the aging of three-month-old triple-transgenic mice, suggesting the therapeutic potential of Zfra in AD prevention. Finally, we demonstrated the presence of pT12-WWOX as aggregates in the cortex of 11-month-old heterozygous *Wwox* mice. The functional role of pT12-WWOX aggregates in vivo is discussed.

## 2. Results

### 2.1. Protein Aggregation in Knockout Wwox^−/−^ Cells and Transforming Growth Factor Beta (TGF-β) Increased the Aggregation

We used our established *Wwox* knockout mice [38,44], as well as wild-type *Wwox*^+/+^, knockout *Wwox*^−/−^, and heterozygous *Wwox*^+/−^ mouse embryonic fibroblasts (MEF) [21,22]. The knockout MEF cells exhibited constitutive phosphorylation of Smad3 of the SMAD pathway and p38 in the MAPK pathway [31]. Compared to the wild-type MEF cells, the *Wwox* knockout cells exhibited intracellularly elevated levels of aggregated TIAF1, whereas the levels of wild-type TPC6A and JNK1 remained relatively unchanged (Figure 1A). When these knockout cells were treated with TGF-β for 1 h, rapid increases in the accumulation of TPC6A, TIAF1, and JNK1 were observed (Figure 1A). The levels of neurofibrillary tangles (NFT) were also elevated in the *Wwox* knockout MEF cells, and tumor necrosis factor alpha (TNF–α) did not increase the NFT expression (Figure 1B). Wild-type TPC6A underwent polymerization in both the cytoplasm and nucleus of the *Wwox* knockout MEF cells, and TGF–β2 increased the polymerization during treatment for 1 h (Figure 1C).

### 2.2. Neurotoxin MPP+ Upregulated TPC6AΔ for Aggregation in Neuroblastoma Cells

Downregulation of WWOX in middle age leads to aggregation of a cascade of proteins, and TPC6AΔ is the first protein to become polymerized or aggregated, which may lead to AD at old age of greater than 70 to 110 [43,44,46]. Under physiological conditions, WWOX binds TPC6AΔ via its *C*-terminal D3 tail, so as to prevent TPC6AΔ self-aggregation [44,46,47]. When WWOX becomes Tyr33 phosphorylated, the self-folded WWOX, due to the binding of intramolecular first WW domain with SDR domain, opens up, so that TPC6AΔ interacts with the fist WW domain [21]. This further prevents TPC6AΔ aggregation. Nonetheless, the higher the MPP+ concentration, the more TPC6AΔ becomes aggregated.

Neuroblastoma SK-N-SH cells were treated with various doses of MPP+ in vitro for 24 h at 37 °C. By non-reducing SDS-PAGE and then Western blot analysis, protein dimerization and further polymerization were observed. The SH3GLB2 dimer showed a dose-dependent increment (Figure 2A). Similarly, TIAF1 polymerization increased upon MPP+ treatment from 125 to 250 µM (Figure 2B). TPC6A∆ showed polymerization at 62.5 µM MPP+ treatment, followed by gradual increase in polymerization at 125 and 250 µM MPP+ treatment (Figure 2C). TPC6A∆ phosphorylated at Ser35 upon treatment with 125 and 250 µM MPP+ (Figure 2D). The level of α-tubulin as a loading control is shown (Figure 2E). In addition, both time-course and dose–response experiments were carried out (Appendix A). MPP+ had no apparent effect on APP expression (Appendix A). However, MPP+ significantly suppressed the expression of pY33-WWOX, and yet significantly increased the expression of Aβ (Appendix A), Overall, our data revealed that TPC6A∆ is most responsive to polymerization caused by MPP+. 

### 2.3. Zfra Restored the Learning and Memory Capabilities in 10-Month-Old 3xTg AD Mice 

Next, we examined the effect of the Zfra4-10 peptide in restoring memory loss in aging 3xTg mice. Ten-month-old triple-transgenic mice (3xTg) for Alzheimer’s disease (AD), possessing mutant genes *Psenl* (M146V), *APPSwe*, and *Tau* (P301L), were used [5]. 3xTg mice received tail vein injections of PBS (sham control) or Zfra4-10 peptide (2 mM in 100 μL PBS) once per week for 4 consecutive weeks, followed by resting for another week. In agreement with our previous report [5], Zfra4-10-treated mice exhibited significant improvement in the novel object recognition task (ORT) (Figure 3A), which is supported by the hippocampus-dependent nonspatial learning and memory.

The abilities of hippocampus-dependent nonspatial memory are expressed as the percentages of novel object exploring time. That is, mice spent their time exploring both objects (time spent on novel object/time spent on both objects), defined as acquisition. During the 5 min acquisition phase, both mouse groups spent equal exploring times on each object (Figure 3A), suggesting that the task environment did not provide preferences for the mice. 

In the short-term memory task (2 h after the acquisition phase), Zfra-treated mice exhibited significant improvement in expanding the exploring times of a replaced new or novel object (Figure 3A). Similarly, in the long-term memory task (24 h after the acquisition phase), the novel object exploring times for the Zfra-treated group were significantly higher than the sham control group (Figure 3A). These observations suggest that the hippocampus-dependent nonspatial memory of 3xTg mice was significantly improved by Zfra treatment. 

Two days after the novel object recognition test, all mice were subjected to the Morris water maze for determining the hippocampus-dependent spatial learning and memory. Both control and Zfra groups showed a time-dependent decrease in escape latency (Figure 4B). However, Zfra-treated mice had a significantly reduced time in escaping from the water and spent more time in the target quadrant (Figure 3B; *p* < 0.05, ANOVA). That is, in the probe test with the platform removed (Figure 3B), the Zfra mice spent more time in the targeted quadrant than the PBS controls (Bonferroni’s post hoc test and ANOVA: ** *p* < 0.01, *** *p* <0.001; Zfra vs. control experiments; sham: *n* = 10; control: *n* = 10). Together, in agreement with our previous observations [5], Zfra was potent in restoring memory loss in aging 3xTg mice.

### 2.4. Zfra4-10 Potently Prevented Age-Dependent Memory Loss in 3xTg Mice 

Next, we investigated whether Zfra4-10 rescued aging-induced neurodegeneration in 3xTg mice. To start with, 3-month-old 3xTg mice received Zfra4-10 peptides via tail vein injections once per week for 4 consecutive weeks. After resting for one week, mice were subjected to examination of their capabilities in memory and learning, as determined by novel ORT. The experiment was repeated every 3 months until the mice reached 10 months old. Our data showed that in the acquisition phase, both control and Zfra mouse groups spent approximately equal exploring times on each object at month 3, 6, and 9 of age (Figure 4A). That is, mice had no specific preferences in the task environment. 

In the short-term memory task, the control mice tended to have age-related decline in novel ORT (Figure 5A). However, Zfra-treated mice did not show memory decline in novel ORT (Figure 4A, *p* = 0.06 ANOVA). Moreover, in the long-term memory task, control groups showed increased decline in novel ORT, whereas the Zfra group had increased memory in handling novel ORT (Figure 4A). 

In the Morris water maze task, the spatial learning and memory in a hippocampus-dependent manner was investigated. Compared to the control group, the Zfra-treated mouse group showed significant restoration of memory loss, as reflected by a significant decrease in escape latency and significantly increased time spent in the target quadrant (Figure 4B). 

### 2.5. WWOX Deficiency Led to Protein Aggregation and Memory Loss in Mice

Previously, we reported that *Wwox* heterozygous mice exhibited an age-related faster decline in both short- and long-term memories than those in 3xTg mice, as determined by novel object recognition tests [5]. To further validate the role of WWOX in memory maintenance, 10-month-old wild-type and heterozygous *Wwox* mice were subjected to behavioral tests. In the novel object recognition task, the non-spatial learning and memory were significantly better in the wild-type than the heterozygous *Wwox* mice (Figure 5A). Again, there was no preference on the acquisition of the designated objects. In both short- and long-term memory analyses, wild-type mice performed significantly better than heterozygous *Wwox* mice, suggesting that WWOX plays a critical role in the maintenance of non-spatial and spatial memory (Figure 5A,B), that is, lacking one allele of *Wwox* gene in the gnome leads to memory deficiency. In parallel with the behavioral analyses, the 10-month-old wild-type mice had significantly reduced amounts of TPC6AΔ in the hippocampus compared to the heterozygous *Wwox* mice, as determined by immunohistochemistry (Figure 5C). 

### 2.6. The Presence of TPC6AΔ, pS35-TPC6AΔ, and SH3GLB2 Aggregates in the Cortex of 11-Month-Old Heterozygous Wwox Mice

We reported the presence of TPC6AΔ and TIAF1 protein aggregates in the brain of middle-aged normal individuals due, in part, to downregulation of WWOX [5,42,44,46]. The protein aggregates continue to stay and accumulate in the human brain without apparent degradation from middle age. These may account for the problematic neurodegeneration when one reaches ages greater than 75 years old. Additionally, the presence of both TPC6AΔ and TIAF1 aggregates has been identified in 3-week-old *Wwox* knockout mice [5], indicating that lack of WWOX rapidly causes protein aggregation in the brain. 

Isoform TPC6AΔ protein aggregated significantly greater in the hippocampus and cortex of 11-month-old heterozygous *Wwox* mice than wild-type mice (Figure 5C). Shown in Figure 6 is the indicated protein aggregation in 11-month-old heterozygous *Wwox* mice and wild-type mice. Significant increases in protein aggregates such as pS35-TPC6AΔ (Figure 6A) and SH3GLB2 (Figure 6C) were observed in the cortex of 11-month-old heterozygous mice. The digitally enlarged Figure 6 are shown in Appendix A. Aggregation of pS37-TIAF1, TIAF1, wild-type TPC6A, and pT181-Tau were not significantly increased (Figure 6B–F). The failure in significant increases was probably due to the presence of a functional *Wwox* allele in the heterozygous mice, which retards protein aggregation. Wild-type TPC6A expression was expressed intracellularly in the neurons of the cortex of heterozygous *Wwox* mice and wild-type mice (Figure 6B).

### 2.7. Identification of pT12-WWOX as Protein Aggregates in the Cortex of Old Heterozygous Wwox Mice

Increased phosphorylation of WWOX at Y33 and Y287 was observed in the cortex of heterozygous *Wwox* mice, as reflected by increased immunogenicity for these proteins (Figure 7A,B). pY287-WWOX appeared to be located in the nuclei, and pY33-WWOX was released into the extracellular matrix. Aggregation for both proteins was poor. Y287 phosphorylation is needed for WWOX to undergo ubiquitination and degradation [20], whereas Y33-phosphorylated WWOX maintains mitochondrial functional and induces apoptosis when overexpressed [9,12,15].

In stark contrast, pT12-WWOX aggregates or plaques were shown in the cortex of 11-month-old heterozygous *Wwox* mice, but not in the wild-type mice (Figure 7C). The data suggest that increased phosphorylation at T12 is sufficient to cause WWOX to undergo aggregation. Under similar conditions, the expression levels of pS14-WWOX were low in both wild-type and heterozygous mice (Figure 7D). Without using a specific primary antibody, negative control staining was shown (Figure 7E). The quality of our newly generated pT12-WWOX antibody, as well as pS14-WWOX antibody [19], is shown (Appendix A). The immunoblot was from immunodeficient NOD-SCID mouse brains. The levels of pS14-WWOX were lower than that of pT12-WWOX.

## 3. Discussion

In this study, we determined that Zfra4-10 peptide is potent in preventing the progression of AD-like symptoms in 3xTg mice. We previously reported that Zfra4-10 peptide restores the memory deficit and mitigates the AD-like symptoms in 3xTg mice [5]. The underlying mechanism is that Zfra blocks the aggregation of pS35-TPC6AΔ, TIAF1, SH3GLB2, tau, and amyloid β and inhibits NF-κB-activation-mediated inflammation [5]. Zfra accelerates protein degradation with a ubiquitin/proteasome-independent mechanism [5]. Zfra4-10 peptide can be used for both therapy and prevention of age-related AD. Indeed, a clinical trial for Zfra in has been planned in the near future. This will cover both AD and cancer. 

Zfra4-10 and Zfra1-31 peptides are potent inhibitors for cancer growth [52,53,54]. When mice receive Zfra peptides, these mice become resistant to the growth of inoculated cancer cells [52]. The underlying mechanism is that Zfra binds membrane Hyal-2 to initiate the signaling of Hyal-2/WWOX/Smad4 [52,53,54]. Overly expressed Smad4 induces cell death by apoptosis. Alternatively, Zfra stimulates the activation of Hyal-2+ CD3- CD19- Z cells in the spleen. Activated Z cells are potent in killing cancer cells both in vivo and in vitro [52,53,54]. Whether activated Z cells mitigate AD symptoms is being investigated. Notably, Zfra1-31 is potent in treating hyperglycemia/diabetes-associated neurodegeneration [29].

TPC6AΔ is the first one among the known three initiators in the protein aggregation cascade [44,45,46,47,48,49,50]. When intracellular WWOX is low or becomes dysfunctional, aggregation of TPC6AΔ starts to occur, followed by polymerization of TIAF1 and SH3GLB2 [44,45,46,47,48,49,50]. The aggregation complex repeats the polymerization or aggregation process and thereby allows accumulation of more protein aggregates in the brain, so as to cause caspase activation, cytochrome c release, and APP degradation for generation of Aβ [44,46,47]. The protein aggregation cascade starts in middle age and may last more than 30 years. By treating neuroblastoma SK-N-SH cells with neurotoxin MPP+, upregulation and aggregation of TPC6AΔ, along with aggregation of TIAF1, SH3GLB2, Aβ, and tau occurred. MPP+ is an inducer of PD, suggesting TPC6AΔ is a common initiator for AD and PD pathogenesis.

Zfra, a 31-amino-acid zinc finger-like WWOX-binding protein, restores memory deficits in 9-month-old 3xTg mice by blocking the aggregation of TPC6AΔ, SH3GLB2, tau, and amyloid β; accelerating intracellular protein degradation; and inhibiting inflammatory NF-κB activation [5]. Whether Zfra-mediated activation of Hyal-2/WWOX/Smad4 signaling is needed for retardation of AD and PD progression remains to be established. Presumably, the signaling may enhance intracellular protein degradation and thereby facilitate neuronal survival. The Hyal-2/WWOX/Smad4 signaling is also crucial for the activation of Hyal-2+ Z cells [52]. Whether activated Z cells block neurodegeneration via secreted cytokines is unknown and remains to be established. Activated Z cells kill cancer cells via unknown cytokines or through direct contact with cancer cells [53,54]. 

Zfra binds WWOX to the *N*- and *C*-terminal domains of WWOX, and this may lead to acceleration of the degradation of WWOX [5,47]. Zfra4-10 or WWOX7-21 peptide alone blocks cancer growth in mice [52]. In contrast, when both peptides are combined in treating cancer, cancer growth cannot be blocked in vivo. Functional nullification is probably due to covalent binding between Zfra4-10 and WWOX7-21. Notably, the stronger the binding of WWOX with intracellular protein partners, the better the cancer suppression [53]. By the same token, neurodegeneration is likely to be retarded when WWOX has strong physical binding interactions with protein partners in the neurons. This likely scenario is very promising and worth pursuing for further investigation.

A recent report showed that high glucose induces the activation of WWOX for causing mitochondrial apoptosis [29]. Zfra blocks the effect of high glucose due to its interaction with WWOX [29]. Zfra can be of therapeutic efficacy in treating hyperglycemia/diabetes-associated neurodegeneration [29]. This finding is important and yet not surprising for Zfra. We believe that Zfra can be a universal drug for targeting many diseases via Hyal-2/WWOX/Smad4 signaling.

The secondary WW domain (WW2) exhibits three antiparallel β-sheets in the solution [55]. However, the tertiary structure of the first WW domain (WW1) is unknown. It is not clear whether these β-sheets are able to stack up together to generate protein aggregates. We determined that WW domains or SDR domain undergo domain self-binding or inter-domain binding [21]. Whether inter-WW domain binding is Y33 phosphorylation dependent remains to be established [21]. 

Our data from immunohistochemistry showed that there was no apparent aggregate formation for pS14-, pY33-, and pY287-WWOX. The presence of pT12-WWOX aggregates in the cortex of heterozygous *Wwox* mice, but not in wild-type mice, suggests that reduced amounts of WWOX in the cytoplasm leads to T12 phosphorylation and increased WWOX protein instability. While pS14-WWOX supports the pathogenic progression toward AD severity and cancer growth and metastasis [5,52,55], the relationship between pT12 and pS14 in WWOX needs further elucidation, so as to increase our understanding of protein aggregation and the downstream TPC6AΔ in the protein aggregation cascade. That is, whether pT12-WWOX aggregates induce TPC6AΔ polymerization remains to be established. Another intriguing issue is that pY33-WWOX is apparently released from the nuclei of neurons to the extracellular matrix in the cortex. pY33-WWOX is critical in maintaining cellular functions and stabilizing the normal physiology of mitochondria with p53 [9]. If this holds true, the way in which accumulation of pY33-WWOX in the extracellular space occurs remains to be established. 

Neurotoxin 1-methyl-4-phenyl-1,2,3,6-terahydropyridine (MTPT) is commonly utilized to investigate the molecular cascade of cell death in dopaminergic neurons [55]. MTPT is less toxic in vivo and is converted to 1-methyl-1-4-phenylpyridinium (MPP+) to cause dopaminergic neuronal degeneration and death [56,57]. MPP+ inhibits the mitochondrial respiration in dopaminergic neurons by suppressing the complex I of the electron transport chain, which results in ATP depletion and increased reactive oxidative species (ROS) [58,59]. MPP+ activates WWOX via Y33 phosphorylation in the rat brain to induce neuronal death [60]. A Tyr33-phosphorylated WWOX peptide (pY33WW; amino acid #28 to 38; KDGWVpYYANHT) was made and was shown to block neuronal death in the rat brain induced by MPP+ [61]. Without y33 phosphorylation, the peptide fails to block neuronal death caused by MPP+. Conceivably, the pY33WW peptide blocks the function of the endogenous full-length pY33-WWOX in the rat brain. pY33-WWOX becomes proapoptotic when it is overexpressed [59,60,61,62].

Under physiological conditions, WWOX binds TPC6AΔ via its C-terminal D3 tail, so as to prevent self-aggregation of TPC6AΔ [46]. When WWOX becomes Tyr33 phosphorylated, the self-folded WWOX from the binding of intramolecular WW domains with SDR domain opens up, so that TPC6AΔ interacts with the WW domains. This further prevents TPC6AΔ aggregation. Nonetheless, the higher the MPP+ concentration, the more TPC6AΔ becomes aggregated.

An additional mechanism is that the Zfra peptide regulates the expression of WWOX. WWOX is a potent inhibitor for GSK3β, due to its binding of GSK3β via the SDR domain. Importantly, WWOX inhibits GSK3β-stimulated S396 and S404 phosphorylation within the microtubule-binding domains of Tau, indicating that WWOX limits GSK3β activity in cells [60,62]. 

## 4. Materials and Methods

### 4.1. Wwox Knockout Mice and Cell Lines

Human neuroblastoma SK-N-SH cells from the American Type Culture Collection (ATCC, Manassas, VA, USA) were cultured in Dulbecco’s modified Eagle’s medium (DMEM) supplemented with 10% fetal bovine serum (FBS) (GIBCO, Carlsbad, CA, USA). 

### 4.2. Synthetic Peptides 

The Zfra4–10 (RRSSSCK) peptide was synthesized by Genemed Synthesis (San Antonio, TX, USA). The peptide was prepared at 10 mM in degassed sterile Milli-Q water as stocks. Each tube was flushed with nitrogen and stored in a −80 °C freezer. For tail vein injections, the peptide stock solution was freshly diluted to 1–4 mM in 100 μL degassed phosphate-buffered saline (PBS). 

### 4.3. Antibodies and Western Blotting Analysis

Commercial antibodies used were against the following proteins: (1) monoclonal α-tubulin, APP, and WWOX (D-9) from Santa Cruz Laboratory; (2) polyclonal Aβ (H-43) from Santa Cruz Laboratory, Santa Cruz, CA, USA; (3) monoclonal pT181-tau and NFT (Biosource, Blue Bell, PA, USA). Homemade antibodies were made in rabbits by using an approved protocol for rabbit use from the IACUC of the National Cheng Kung University Medical College. Antibodies were produced against the following proteins: TIAF1 [38,39,43], pS37-TIAF1 [38,39,43], TPC6A [5,41,43], TPC6AΔ [5,41,43], pS35-TPC6AΔ [5,41,43], SH3GLB2 [5], WWOX [5,6,8,12], pS14-WWOX [19,49], and pY33-WWOX [6,8,12]. In addition, we generated an antibody against pT12-WWOX7-21 using the synthetic peptide NH-AGLDDpTDSEDELPPG-COOH. The generated antibody was characterized by Western blotting and peptide blocking, as previously described [61]. For Western blotting, cells were lysed with a lysis buffer (1% SDS, 0.5% NP-40, 0.1% Triton X-100, in the presence of protease inhibitor cocktail (sigma) and 1mM PMSF and 1mM Na_3_VO_4_) on ice for 30 to 60 min. The lysates were centrifuged at 4°C (13,200 rpm, 15 min) and then processed by standard SDS-PAGE and immunoblotting.

### 4.4. Animals

We followed the National Institutes of Health Guidelines for animal research (Guide for the Care and Use of Laboratory Animal) and the approved protocol by the National Cheng Kung University Institutional Animal Care and Use Committee. The 3xTg mice (B6; 129-Psen1tm1Mpm Tg[APPSwe, tauP301L]1Lfa/Mmjax) were from the Jackson Laboratory (Bar Harbor, ME, USA) [5,63,64]. The mice were housed in automatically conditioned rooms (temperature 23 ± 1 °C, humidity 55 ± 5%, 12 h light/12 h dark cycle, light cycle beginning at 06:00), which are located in the Laboratory Animal Center of National Cheng Kung University (Tainan, Taiwan). These mice had unrestricted access to food and water [5]. Genotyping was carried out for all mice using a protocol provided by the Jackson Laboratory. 

For Zfra-mediated restoration of memory loss and learning deficit, five 10-month-old 3xTg mice (body weight: 31~35 g) received four consecutive injections of Zfra4–10 solution (2 mM in PBS, 100 μL each injection) via the tail veins on a weekly basis. Under similar conditions, the sham mouse group received PBS injections only. One week later, behavioral tests were carried out, and then mice were sacrificed by CO_2_ anesthesia (75% CO_2_/25% O_2_) and perfused with 10 mL PBS (to reduce background in tissue section staining with specific antibodies). Brains were then rapidly harvested.

To investigate whether Zfra prevented age-related neurodegeneration, 3-month-old 3xTg mice received Zfra4-10 peptide or PBS via tail vein injections once per week for 3 consecutive weeks. One week later, mice were subjected to behavioral tests. Zfra treatment was repeated twice when mice reached 6 and 9 months old. Again, behavioral tests were then carried out.

### 4.5. Novel Object Recognition Test 

Normally, the experiment was carried out from 6 PM each day as previously described [5,63]. After Zfra or PBS treatment, mice were allowed to rest for a week. Mice were initially habituated to a polycarbonate box (47.5 cm × 25.8 cm × 21 cm) by staying in there for 10 min per day for 3 consecutive days. Twenty-four hours later, each mouse faced two identical objects (glass bottle, 4 cm diameter and 6.5 cm high) positioned 7 cm apart away from a wall. The durations that the mouse explored each object were recorded over 5 min. For the short-term memory (STM) test, the mouse was placed into the box 2 h later, facing an old and a new object (white Lego bricks, 3 × 3 × 6 cm). For the long-term memory (LTM) test, the mouse was allowed in the box 24 h later to face an old and a new object (white plastic bottle, 3.5 cm diameter and 7.2 cm high). The times spent in exploring each object during the 5 min period were recorded in either the STM or the LTM test. All of the objects were cleaned by 70% alcohol between trials to reduce olfactory cues.

### 4.6. Morris Water Maze Test

The Morris water maze test was intended to measure the hippocampus-dependent spatial learning and memory, which was carried out 1 day after the novel object recognition test [5]. The Morris water maze was performed in a custom-made circular pool with a diameter of 120 cm and a wall height of 31 cm. The pool was filled with clear tap water at a temperature of 24 ± 1°C and depth of 25 cm. The 4-day training session started at 6 pm. Each session consisted of four swim trials (maximum 120 s per trial) with different quadrant starting positions for each trial. On the morning of the fifth day, mice were subjected to a probe test. The mouse was placed in the pool in the southwest position, the longest distance away from the previous platform position (northeast). The mouse was allowed to swim for 60 s without a platform present. A CCD camera was used to record the whole process, and the escape latency (i.e., time to reach the platform, in seconds), path length, and swim speed (cm/s) were analyzed by a EthoVision video tracking system (Noldus Information Technology, Wageningen, The Netherlands).

### 4.7. Brain Sections and Immunohistochemistry

After the experiments, mice were sacrificed. Brains were harvested and fixed with 4% buffered paraformaldehyde. Paraffin-embedded blocks were made. Tissue sectioning and immunohistochemistry were carried out as described previously [5,7,12]. Frontal sections (6 μm in thickness) were made and then de-paraffinized by xylene, followed by stepwise hydration using decreasing concentrations of ethanol. Antigen retrieval was performed using Tris-EDTA (pH 8.0) with 0.5% Tween 20 by boiling for 15 min using a microwave. The tissue section was blocked with 1% H_2_O_2_ in Tris-buffered saline (TBS) for 15 min to remove the peroxidase background, followed by washing 3 times for 5 min each using TBS with 0.5% Tween 20. Sections were then blocked with 0.5% Tween 20 and 5% BSA at room temperature (RT) for at least 1 h. Immunohistochemistry was carried out using an assay kit (HRP-conjugated secondary antibody (DAKO)). Specific primary antibodies are listed in the aforementioned Section 4.3. Microscopy was carried out, and 10 pictures per microscopic field were taken (OLYMPUS BX51 Microscope). Images were digitally analyzed and quantified using ImageJ software. 

### 4.8. Statistical Analysi

Where indicated, all data (mean ± standard error) were analyzed by a two-tailed Mann–Whitney test (e.g., a novel object recognition test and a probe test of the Morris water maze) and a two-tailed unpaired *t*-test (protein expression in immunohistochemistry) [5]. Bonferroni post hoc tests were performed if significant (*p* < 0.05) main effects were found [5].

## 5. Conclusions

The first critical discovery in this study is that the Zfra4-10 peptide effectively retards age-dependent neurodegeneration in 3xTg mice, as determined by an novel object recognition task and Morris water maize analysis, that is, prevention of neurodegeneration at old age can be achieved by receiving a Zfa4-10 peptide at a young age. Zfra suppresses the aggregation of TPC6AΔ, TIAF1, SH3GLB2, and Aβ and significantly downregulates the aggregation of pT181-Tau [5]. Zfra also enhances the degradation of intracellular proteins. Zfra covalently crosslinks with target proteins for proteolytic degradations—the so-called zfration—and thereby prevents the Aβ generation and neurodegeneration shown in AD [5]. 

The second important finding is that as an inducer of PD, neurotoxin MPP+ rapidly increases WWOX phosphorylation at Tyr33 [29] and upregulates the expression of TPC6AΔ, followed by TIAF1, SH3GLB2, Aβ, and tau in neuroblastoma SK-N-SH cells. Upon MPP+ induction, TPC6AΔ is the first protein to polymerize and lead others to aggregation, suggesting that TPC6AΔ is a common initiator for AD and PD pathogenesis. Finally, pT12-WWOX becomes aggregated in the cortex of 11-month-old *Wwox* heterozygous mice, compared to age-matched wild-type mice. Whether aggregation of pT12-WWOX links to TPC6AΔ polymerization is unknown. 

## 6. Patents

Nan-Shan Chang and Yu-Min Kuo. Method for treating Alzheimer’s disease and a method for downregulating protein aggregation in the brain. U.S. patent #10344065. 6 March 2018.

## Figures and Tables

**Figure 1 ijms-23-14510-f001:**
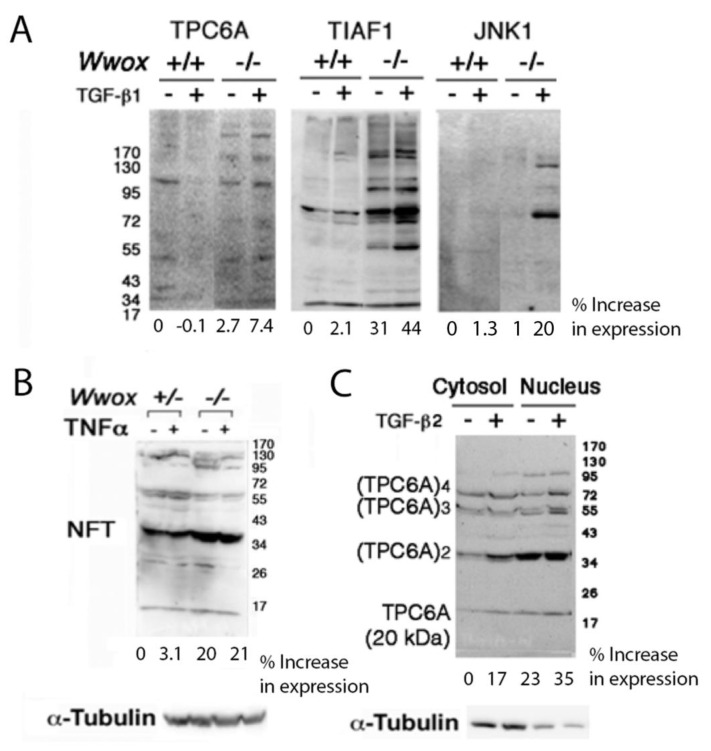
Protein aggregation in *Wwox* knockout MEF cells. (**A**) The expression of endogenous TPC6A, TIAF1, and JNK1 was determined in the wild-type and knockout *Wwox* MEF cells (with gel run under reducing SDS-PAGE and then Western blotting). These cells were treated with TGF-β1 (5 ng/mL) for 1 h. (**B**) The presence of NFT was shown in the heterozygous and knockout *Wwox* MEF cells. TNF-α (100 ng/mL) did not increase NFT expression in both cells during treatment for 1 h. (**C**) TGF-β1 (5 ng/mL) increased polymerization of wild-type TPC6A in the knockout *Wwox* MEF cells during treatment for 1 h at 37 °C.

**Figure 2 ijms-23-14510-f002:**
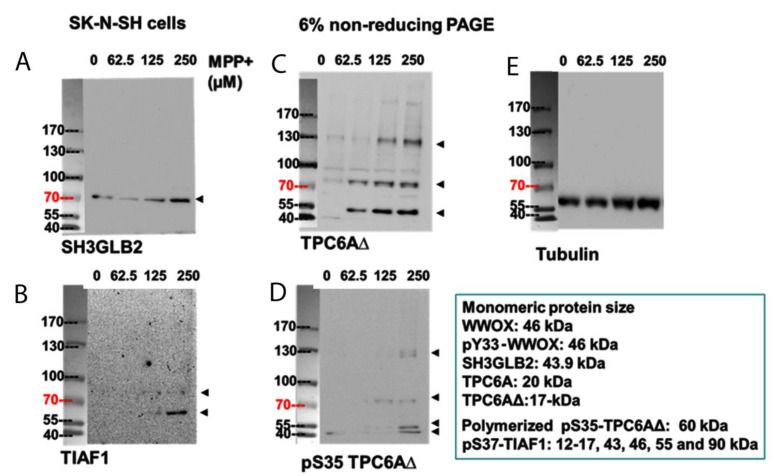
Polymerization of TPC6A∆, TIAF1, and SH3GLB2 under MPP+ stimulation. SK-N-SH cells were seeded to 70% confluence and cultured overnight with 10% fetal bovine serum. Prior to stimulation, cells were synchronized under serum-free conditions for 20 h. Cells were stimulated with MPP+ at the indicated doses for 24 h. By non-reducing SDS-PAGE (6%) and Western blotting, protein polymerization was observed. (**A**) SH3GLB2 dimerized in a dose-dependent manner. (**B**) Polymerized TIAF1 was shown upon increasing MPP+ from 125 to 250 µM. (**C**) TPC6A∆ showed polymerization at 62.5 µM MPP+ treatment, followed by gradual increment of polymerization at 125 and 250 µM MPP+ treatment. (**D**) TPC6A∆ phosphorylated at Ser35 at 125 and 250 µM MPP+ treatment. Two individual repeats were performed for each experiment. Representative figures are shown. (**E**) As an internal standard, the level of α-tubulin is shown.

**Figure 3 ijms-23-14510-f003:**
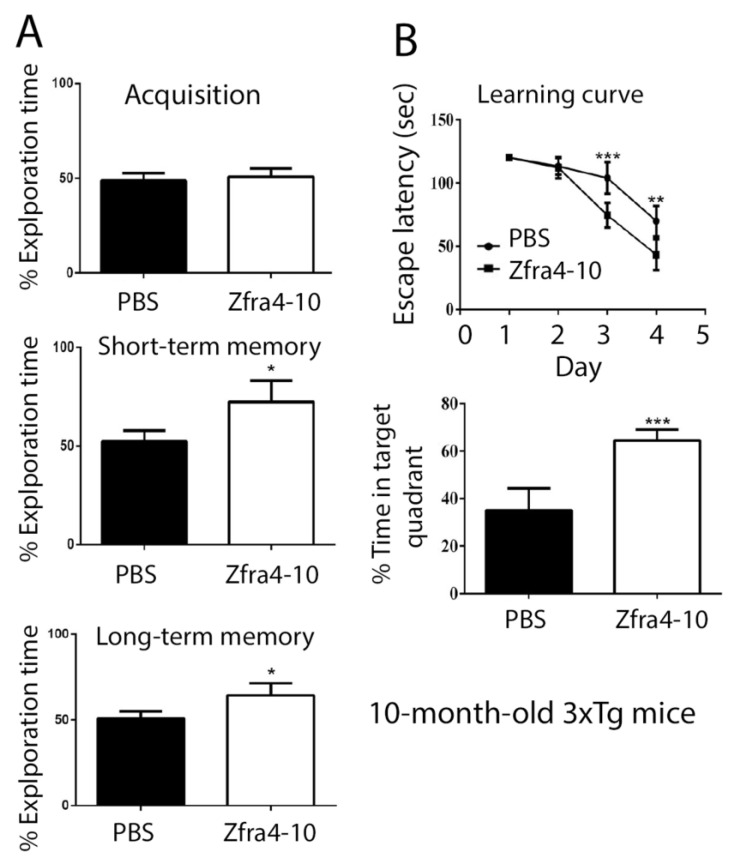
Zfra4-10 peptide restored the learning and memory of 10-month-old 3xTg AD mice. (**A**,**B**) 3xTg mice, aged 10 months old, received Zfra4-10 tail vein injections, followed by examination of their learning and memory capabilities by a novel object recognition task (**A**) and a Morris water maze (**B**) at 1 month after the injections. The percentages of novel object exploring time (time spent on novel object/time spent on both objects) of acquisition, short-term (2 h) delay, and long-term (24 h) delay are shown (control group: sham, *n* = 10; Zfra injection group: Zfra4-10, *n* = 10). In the Morris water maze, the learning curve represents the latency of finding the hidden platform during the training session. The probe test was quantified as time spent in the target quadrant (Bonferroni’s post hoc test and ANOVA: * 0.01< *p* < 0.05, ** 0.001< *p* < 0.01, *** *p* < 0.001, experimental group vs. respective control group. Sham: *n* = 10; control: *n* = 10).

**Figure 4 ijms-23-14510-f004:**
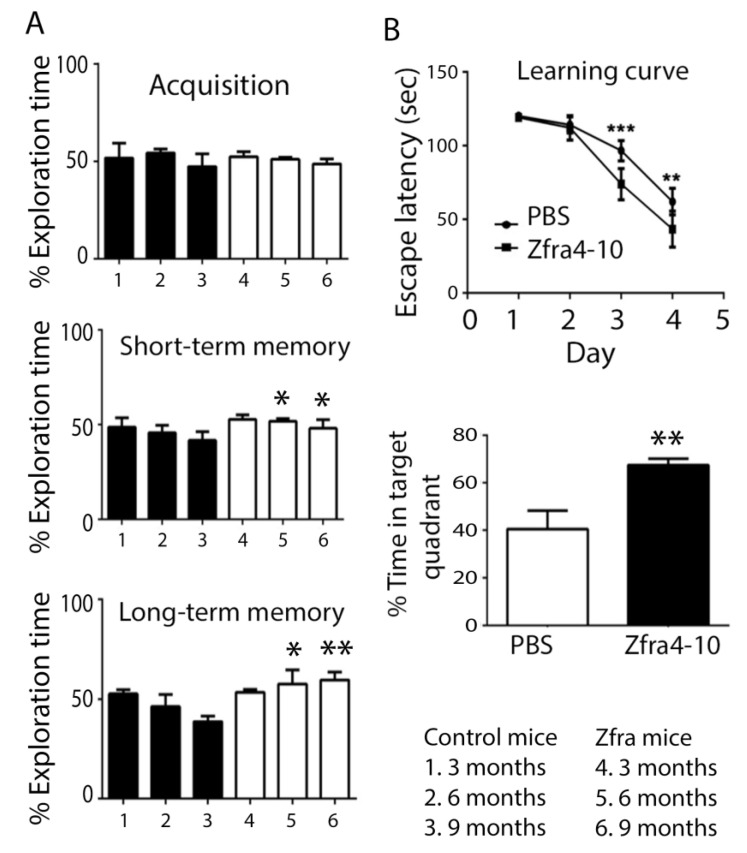
Zfra4-10 blocked age-dependent memory loss in 3xTg mice from 3 to 9 months of age. (**A**,**B**) Three-month-old 3xTg mice received Zfra4-10 peptide via tail vein injections once per week for 3 consecutive weeks, followed by one-week rest, and then they were subjected to novel ORT (**A**) and Morris water maze analysis (**B**). The animal behavior experiments were repeated when mice reached 6 and 9 months old, respectively. The percentages of novel object exploring time of acquisition, short-term (2 h) delay, and long-term (24 h) delay are shown (*n* = 5 for each group). The learning curve represents the escape latency to find the hidden platform during a training session. The probe test was quantified as time spent in the target quadrant (Bonferroni’s post hoc test and ANOVA: * *p* < 0.05, ** *p* < 0.01, *** *p* < 0.001 vs. the respective control group; *n* = 5).

**Figure 5 ijms-23-14510-f005:**
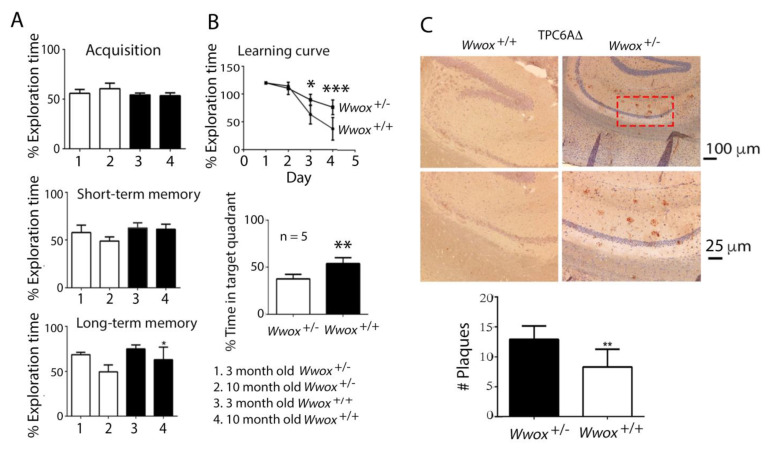
WWOX expression was critically associated with memory maintenance. (**A**,**B**) Wild-type and heterozygous *Wwox* mice at age 3 and 10 months were subjected to learning and memory tests by the novel object recognition task (**A**) and Morris water maze (**B**). The percentages of novel object exploring time of acquisition, short-term (2 h) delay, and long-term (24 h) delay are shown (*n* = 5). The learning curve represents the latency to find the hidden platform during a training session. The probe test was quantified as time spent in the target quadrant (Bonferroni’s post hoc test and ANOVA: * *p* < 0.05, ** *p* < 0.01, *** *p* < 0.001 vs. respective control group; *n* = 20). (**C**) Micrographs of 11-month-old wild-type and heterozygous hippocampal areas. TPC6AΔ plaques were significantly upregulated in the heterozygous mice (** *p* < 0.01 vs. respective control group; *n* = 20). The number of plaques per microscopic field at 200× is shown.

**Figure 6 ijms-23-14510-f006:**
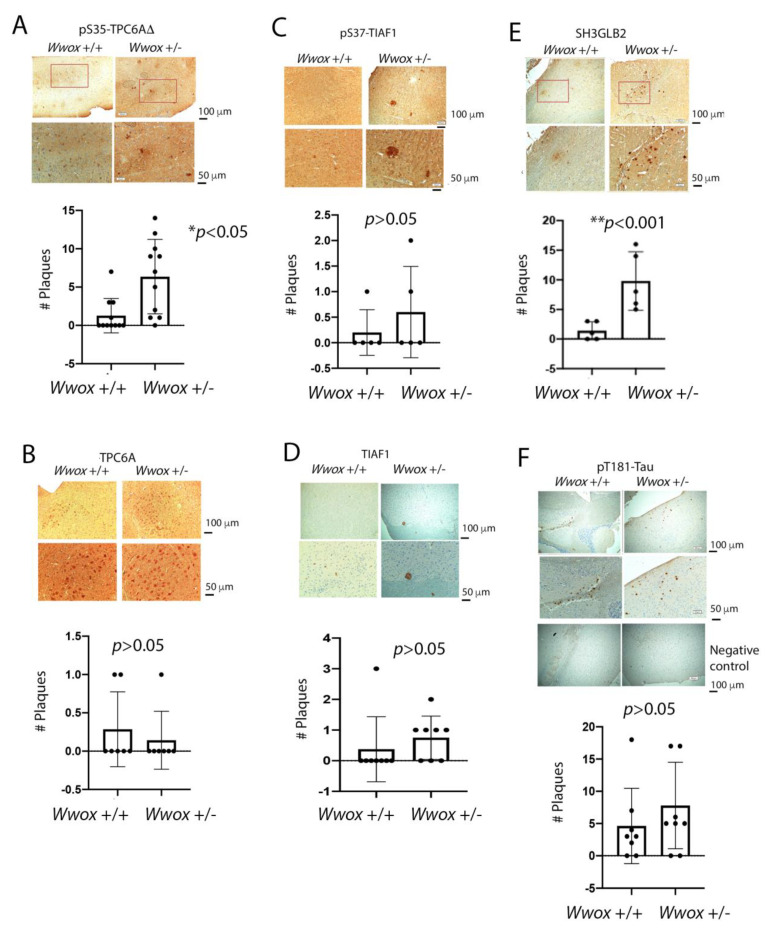
Presence of pS35-TPC6AΔ and SH3GLB2 aggregates in the cortex of 11-month-old heterozygous *Wwox* mice. (**A**–**F**) By immunohistochemistry, protein expression of pS35-TPC6AΔ (**A**), TPC6A (wild type) (**B**), pS37-TIAF1 (**C**), TIAF1 (**D**), SH3GLB2 (**E**), and pT181-Tau (**F**) was examined in the brain cortex of 11-month-old heterozygous *Wwox* mice and age-matched wild-type mice. The extent of protein aggregation was examined (*n* = 5 to 10; Student’s *t*-test). Negative controls without staining with primary antibodies are shown. The digitally enlarged pictures are shown in Appendix A.

**Figure 7 ijms-23-14510-f007:**
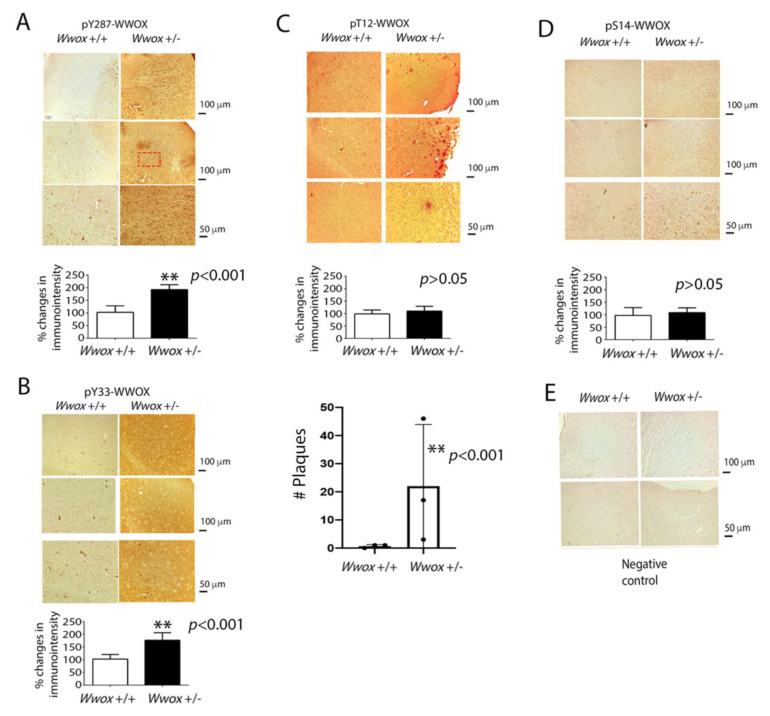
Identification of pT12-WWOX as aggregates in the brain cortex of 11-month-old heterozygous *Wwox* mice. (**A**–**E**) Compared to the wild-type mice, increased immunointensity of staining was shown in the cortex of heterozygous *Wwox* mice using an antibody against pY287-WWOX (**A**) and pY33-WWOX (**B**). The presence of pT12-WWOX aggregates (**C**), but not pS14-WWOX (**D**), was shown in the cortex of heterozygous *Wwox* mice. No pT12-WWOX aggregates were found in the wild-type mice. The bar graphs show mean ± standard deviation (*n* = 5; ** *p* < 0.001, experimental group vs. respective control group, Student’s *t*-tests). The number of plaques per microscopic field (100x magnification) is shown for pT12-WWOX (*n* = 3) (**C**). No plaques are shown with pS14-, pY33-, and pY287-WWOX. (**E**) Negative controls without staining with primary antibodies are shown. The digitally enlarged pictures are found in Appendix A.

## Data Availability

The corresponding author will provide the data that support the findings of this study, which are available upon reasonable request.

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
