# Peer review of "Zfra Inhibits the TRAPPC6AΔ-Initiated Pathway of Neurodegeneration"

_ijms, 2022, doi:10.3390/ijms232314510_

Round 1

Reviewer 1 Report

I would like to thank the Authors of the present Manuscript for the opportunity to provide commentary on their work.

The Authors have indeed compiled a huge amount of information in this paper, spanning from gel electrophoresis to immunohistochemistry assays, to behavioral tests on triple-mutant mice and for several gene products as well as active peptides. Overall, the manuscript is well-written and sufficiently well-structured, although I would appreciate the discussion to be a single chapter, and not a series of paragraph with their own title. The content itself is of high quality, and I have no notes about the methodology, which is clearly described.

Notes:

1) the Title is too complex and wordy, I would advise simplifying it, also considering that the results shown in the paper also involve a number of other genes

2) the amount of information makes the text extremely complex, in terms of gene names, mutants, and general nomenclature. This does not detract from the form of the text itself, which is to praise.

3) several Figures and their related text are not completely clear. Specifically, Figures 2 and 7 look very cluttered and the single images are sometimes difficult to read.

Author Response

Dear Reviewer:

Thank you so much for your enthusiasm and outstanding contribution. We really appreciate your comments. 

1The Authors have indeed compiled a huge amount of information in this paper, spanning from gel electrophoresis to immunohistochemistry assays, to behavioral tests on triple-mutant mice and for several gene products as well as active peptides.

Answer:  Thank you.

2Overall, the manuscript is well-written and sufficiently well-structured, although I would appreciate the discussion to be a single chapter, and not a series of paragraph with their own title. The content itself is of high quality, and I have no notes about the methodology, which is clearly described.

Answer:  Thank you. As requested, the Discussion is now in a single chapter.

3) Title: Regarding " the Title is too complex and wordy, I would advise simplifying it, also considering that the results shown in the paper also involve a number of other genes".

Answer:  As per your request, the Title has been changed to "Zfra inhibits TRAPPC6AΔ-initiated pathway of neurodegeneration". For additional proteins, please refer to data in the new Figure 1 and supplementary Figure 1 and 2. We have done gene expression profiling analysis using mouse brains. These mice were treated with or without Zfra. More work needs to be done to validate how Zfra actually blocks and prevents neurodegeneration in 3xTg mice.

4) Simplicity: Regarding "the amount of information makes the text extremely complex, in terms of gene names, mutants, and general nomenclature. This does not detract from the form of the text itself, which is to praise".

Answer:  Thank you. We apologize for the oversized manuscript. As requested, the text has been thoroughly revised, trimmed and polished. In the Introduction, we have made it reader friendly, as many neuroscientists are not familiar with WWOX and TRAPPC6A. Also, we have deleted the original Figures 2 and 5. 

a) Deletion of the original Figure 2: MPP+ induces secretion of proteins to the culture supernatants and ECM. More studies are being conducted. This will allow us to make a better understanding of how secreted proteins in promoting or blocking AD progression.

b) Deletion of the original Figure 5: We have accidentally made a mistake by having published graphs added in the text. Our apologies. Indeed, we have updated the statistics in this figure. To facilitate future study, we have decided to delete.

5) Figure legends: several Figures and their related text are not completely clear. Specifically, Figures 2 and 7 look very cluttered and the single images are sometimes difficult to read.

Answer:  Thank you. As requested, we have thoroughly revised the legend for the original Figure 7 (now Figure 6). As mentioned above, Figure 2 has been deleted. Other needed areas have also been thoroughly revised.

6) Additional changes

A new figure 1 has been added. This figure deals with protein aggregation in Wwox knockout MEF cells, compared with wild type MEF cells. We also tested the effect of cytokines such as TGF-β and TNF-α.

In the Supplementary Figures 1 and 2, we have added the time-course and dose-response experiments to show the effect of MPP+ on protein expression in SK-N-SH cells. In principle, we discovered that MPP+ suppresses the expression of pY33-WWOX and TPC6AD in 12 hr, which negatively correlates with upregulation of amyloid β formation.

Reviewer 2 Report

The authors present preliminary data about MPP+ treatment of neuroblastoma SK-N-SH cells on polymerization of TPC6Adelta TIAF1 and SH3GLB2 and secretion of APP, Abeta, Tau, TPC6Adelta and TPC6A. In a second part the authors set out to demonstrate the effects of i.v. injection of a Zfra4-10 peptide (RRSSSCK) on learning and memory and protein aggregation in a 3xTg Alzheimer mouse model. In addition protein aggregates were investigated in brain tissue of heterozygous WWOX knock out mice.

Neither the Western Blot analyses nor the immunohistochemistry analyses suffices scientific standards.
In Figure 1 and Figure 2 a single biological replicate is not sufficient to prove a scientific statement. At least n=4 biological replicates for each condition (control and MPP+ concentrations) with statistical analysis are necessary. Overall the Western Blots are of poor quality, on some blots the monomer ist not shown (Fig. 1 A, B, C, D). The PAGE-gels should show the complete molecular weight range from monomer to oligomers. The blots for TIAF1 should also be done with a phospho-independent TIAF1 antibody, because phospho-psecific antibodies tend to have higher unspecific cross-reactivity.

Parts of Figure 5 are already published in Alzheimers Dement 2017 3(2):189-204 PMID: 29067327. This applies to parts of Figure 5a, 5c and 5d.
This is partially obscured by different cropping and flipping of picture panels. For me this looks like scientific fraud.

A more general problem ist the quality of the immunohistochemistry. The spots labeled as aggregates in Figure 5 and Figure 7 look more like staining artifacts. The pictures miss scale bars and the resolution is too low to see cellular structures. Tau aggregation in human AD and mutated tau animal models occur intracellularly and not as diffuse spots as shown in Figure 5a. Also amyloid plaques shown in Figure 5B are not visible. To proof the nature of amyloidogenic aggregation in both cases a Thioflavin S positive signal should be visible on the same tissue section.

Even 3xTg AD mice don't show Alzheimers disease, they just model some aspects of this human-specific disease. Therefore, the statement in line 100 that "Zfra prevents AD progression" is overexaggerated.

Another point which is suspect with this manuscript is the extent of self-citation. Almost 2/3 of all citations refer to the same group/institution. The authors should check, whether this excessive self-citation is necessary.

I will stop adding more critical points because in my opinion, at the present state this manuscript is not acceptable for publication in any journal.

Author Response

In principle, this reviewer has rejected our article. We will not answer this comments, as advised by a journal editor. However, we appreciate his/her contribution and effort. We have revised according to this reviewer's opinion and have listed in our rebuttal letter.

Reviewer 3 Report

A very good and sound paper. Data are abundant and enough for two research papers. In my humble opinion, it can be published as it is.

Author Response

Dear Reviewer:

Thank you for your outstanding effort, contributions, and acceptance of the manuscript. A general consensus of this article is that it is oversized. We have  trimmed down the article, and meanwhile made changes as follows:

1) Deletion of the original Figure 2: MPP+induces secretion of proteins to the culture supernatants and ECM. More studies are being conducted. This will allow us to make a better understanding of how secreted proteins in promoting or blocking AD progression. 

2) Deletion of the original Figure 5: We have accidentally made a mistake by having published graphs added in the text. Our apologies. Indeed, we have updated the statistics in this figure. To facilitate future study, we have decided to delete.

3) Addition of a new Figure 1: This figure shows protein aggregation in the Wwox knockout MEF cells, and TGF-β increased protein aggregation.

4) As suggested by one of the reviewers, we have shortened the title.